# Enhancing Primary Adherence to Prescribed Medications through an Organized Health Status Assessment-Based Extension of Primary Healthcare Services

**DOI:** 10.3390/ijerph16203797

**Published:** 2019-10-09

**Authors:** Nouh Harsha, Magor Papp, László Kőrösi, Árpád Czifra, Róza Ádány, János Sándor

**Affiliations:** 1Department of Preventive Medicine, Faculty of Public Health, University of Debrecen, Debrecen 4012, Hungary; nouhharsha@gmail.com (N.H.); adany.roza@sph.unideb.hu (R.Á.); 2Doctoral School of Health Sciences, University of Debrecen, Debrecen 4012, Hungary; magorpapp@gmail.com; 3Semmelweis Center for Health Promotion, Medical Faculty, Semmelweis University, Budapest 1094, Hungary; 4Department of Financing, National Health Insurance Fund, Budapest 1139, Hungary; korosi.l@neak.gov.hu

**Keywords:** medication adherence, relative dispensing ratio, primary healthcare, health status assessment, patient–physician cooperation

## Abstract

This study was part of monitoring an intervention aimed at developing a general practitioner cluster (GPC) model of primary healthcare (PHC) and testing its effectiveness in delivering preventive services integrated into the PHC system. The aim was to demonstrate whether GPC operation could increase the percentage of drugs actually dispensed. Using national reference data of the National Health Insurance Fund for each anatomical–therapeutic chemical classification ATC group of drugs, dispensed-to-prescribed ratios standardized (sDPR) for age, sex, and exemption certificate were calculated during the first quarter of 2012 (before-intervention) and the third quarter of 2015 (post-intervention). The after-to-before ratios of the sDPR as the relative dispensing ratio (RDR) were calculated to describe the impact of the intervention program. The general medication adherence increased significantly in the intervention area (RDR = 1.064; 95% confidence interval (CI): 1.054–1.073). The most significant changes were observed for cardiovascular system drugs (RDR = 1.062; 95% CI: 1.048–1.077) and for alimentary tract and metabolism-specific drugs (RDR = 1.072; 95% CI: 1.049–1.097). The integration of preventive services into a PHC without any specific medication adherence-increasing activities is beneficial for medication adherence, especially among patients with cardiovascular, alimentary tract, and metabolic disorders. Monitoring the percentage of drugs actually dispensed is a useful element of PHC-oriented intervention evaluation frames.

## 1. Introduction

Adherence to medications can be defined as taking medications as prescribed by healthcare providers [1,2]. Nonadherence to medications can occur when patients delay or do not fill medication prescriptions, do not take the desired dosage, or decide to discontinue their medications [3]. Nonadherence has been described as a global epidemic [4]. It affects various patient groups, particularly those with chronic health conditions [5,6,7]. Approximately one-third of written prescriptions are not dispensed, and approximately 50% of dispensed medications are not taken as recommended [8,9,10,11,12].

Nonadherence has a complex, multifactorial etiology [13,14,15]. The World Health Organization (WHO) classifies the factors affecting medication adherence [16] as socioeconomic status (e.g., poverty, low level of literacy, lack of social support) [17,18], medical conditions (e.g., symptom severity and disability) [14,19], therapy-related factors (e.g., regimen complexity, duration, and side effects) [8,20], and healthcare system specialties (e.g., patient–physician relationship, provision of services) [21,22]. The weight of this issue is reflected in the recommendation of the WHO to use the percentage of drugs actually dispensed as a basic indicator of patient care [23].

The consequences of nonadherence are numerous. It results in accelerated progression of the disease, avoidable complications, increased hospitalization and disability [24,25,26], reduced productivity [4], a lower quality of life, and elevated mortality [13]. Evidently, it increases healthcare costs enormously [27,28,29,30,31,32,33].

Several simple and complex interventions have been tried to improve adherence. Some of the interventions that have been examined have entailed reducing the number of prescribed drugs and adjusting the dose [27,34], giving medication reminders and improving medication schedules [35], and employing educational strategies [8,36,37]. More complex and comprehensive strategies have also been implemented and tested. Such strategies have included improving the patient–physician relationship, communication, and trust; expanding and emphasizing the pharmacist role in primary healthcare [9,27,38]; increasing awareness of the patient regarding his/her disease and medications [39,40]; controlling medication-related side effects and performing patient habit analyses [26,41]; examining patient motivation, support, and follow-up [8,13,14,42]; and keeping the patient in the core of the healthcare process and considering his/her feedback [43].

Since adherence is a multifactorial issue, experiences with the limited effectiveness of unimodal interventions, which target one aspect of nonadherence, are not surprising [40,44]. Multifaceted, comprehensive approaches with strategies designed and tailored to suit individual patients or groups have been found to be the most effective [12,13,45]. Interventions delivered by a number of health professionals, including physicians, pharmacists, nurses, and community healthcare workers, have been proven to be effective in increasing adherence [46,47,48,49].

Additionally, strategies must be simple enough to put into daily practice [50]. Approaches that focused on patient follow-up have been found to be very effective [51,52]. A key issue is the integration of approaches that encourage adherence into a healthcare system and services [53].

Unfortunately, there is no gold standard for a comprehensive approach that can be used to enhance adherence [54]. One meta-analysis of many interventions indicated an increase in adherence magnitude from 4% to 11% [10].

### Objectives

Our investigation was part of monitoring the “Public Health-Focused Model Program for Organizing Primary Care Services Backed by a Virtual Care Service Center” [55], which is aimed at developing a general practitioner cluster (GPC) model of primary healthcare (PHC) and testing its effectiveness in delivering preventive services integrated into the PHC system. The goal of this model, in line with the recommendations of the WHO [56], was to complete a traditional core PHC team of one GP and one nurse with other health professionals who were formerly not available at the PHC level [57,58]. The GPC is a multimodal PHC intervention based on (1) organized, population-level general health checks implemented by public health experts and nurses to explore unmet needs, (2) follow-ups of at-risk patients by providing care via the extended PHC team, (3) new services and nonmedical activities (counseling by a dietitian and psychologist, treatment by a physiotherapist, supporting patient–GP cooperation with mediators), and (4) supervision by a GP.

Our study aimed to demonstrate whether the GPC operation, which aims to improve the general effectiveness of a PHC, could increase primary adherence (the percentage of drugs actually dispensed), thereby reflecting improvement in the quality of patient–GP collaboration.

## 2. Methods

### 2.1. Setting 

Typical Hungarian primary healthcare teams (composed of one general practitioner and one practice nurse) were invited to form GPCs in 2012. The core team was completed with other health professionals. The GPCs were designed to offer preventive services and health promotion interventions in addition to the traditionally given curative, acute, and emergency services. GPC composition and operation rules are summarized elsewhere in detail [57]. Briefly, six core primary healthcare teams formed one GPC. Each GPC employed 1 dietician, 1 physiotherapist, 1 health psychologist, 1 community nurse, 2 public health specialists, and 12 health mediators. Finally, 4 GPCs were established. New services of the program were launched in 2014 after elaboration on care protocols, the training of staff, and the creation of new infrastructures. All adults over 18 years of age whose GPs were involved in the program were called to participate in an organized health status assessment performed by a public health specialist and a community nurse. Details of the assessment have been described earlier [59,60]. The importance of the risk factors and/or conditions identified by the health status assessment was evaluated in a medical risk assessment carried out by a GP. Further, new services of the program that were not available formerly at the level of PHC included the following: GPs referred patients to lifestyle counseling and treatment sections provided by physiotherapists, dieticians, psychologists, and public health specialists to address risk factors and conditions and improve health knowledge and adherence to medical and health professionals’ advice [61].

### 2.2. Data Sources

Data on the number of drugs prescribed by the GPs and dispensed to patients during the first quarter of 2012 (2012Q1, representing the before-intervention status) and the third quarter of 2015 (2015Q3, representing the post-intervention status) were provided by the National Health Insurance Fund, which is the national institution that contracts with each general medical practice (GMP) of Hungary. The number of prescriptions and the number of dispensed prescriptions were the subjects of the analysis. GMP-specific data were stratified by sex, age group (18–19, 20–24, 25–29, …, 75–79, 80–84, 85–*X* years), and eligibility for an exemption certificate (through which socioeconomically disadvantaged patients who require long-term care can get drugs and medical devices free of charge). This kind of database was prepared for each anatomical–therapeutic chemical classification (ATC) group of drugs [62]. Groups of “antineoplastic and immunomodulating agents” and “antiparasitic products, insecticides, and repellents” were not analyzed because these drugs are not typically prescribed by GPs according to Hungarian rules.

### 2.3. Statistical Analysis

The amount of prescribed and dispensed drugs and the percentage of dispensed prescribed drugs (dispensed-to-prescribed ratio, DPR) were calculated for the aggregated intervention population and for the whole country by ATC group and by sex, age group, and eligibility for an exemption certificate for the 2012Q1 and the 2015Q3 periods. We used the DPR as a key indicator of patient care to assess primary adherence to prescribed medications [23].

The sex-, age-, and exemption certificate-specific number of written prescriptions for the intervention population and national reference DPRs were used to compute the expected number of dispensed prescriptions for the intervention population (by summarizing the expected number of dispensed prescriptions across the strata). The ratio of the observed number of dispensed prescriptions and the total expected number of dispensed prescriptions was computed as the intervention population-specific standardized DPR (sDPR) for each ATC group.

The ratio of 2015Q3-specific and 2012Q1-specific sDPRs as the relative dispensing ratio (RDR) was calculated for each ATC group to describe the impact of the intervention program on the DPR. The statistical evaluation was carried out using a 95% confidence interval (95% CI) of the measures. SPSS version 20 was used to complete the calculation processes.

### 2.4. Patient and Public Involvement

Although patients’ needs and public health importance were investigated in this study, the development of the research question and study design was carried out without involvement of the patients and the public. Because aggregated patient data were processed in the statistical works, the analyses did not require the active contribution of patients or their representatives.

## 3. Results

There were 33,101 adults older than 18 years in the intervention population. The age distribution of the intervention population was similar to the national reference before the intervention, but it deviated from the national reference in a statistically significant manner by the end of the study period. There was a statistically significant, but not important, overrepresentation of males and individuals eligible for an exemption certificate in the intervention population. The reference population of adult Hungarians totaled 7,886,662 and 7,745,112 in the first quarter of 2012 and in the third quarter of 2015, respectively (Table 1).

More than half of this target community (*N* = 18,833, 57.9%) participated in an organized health status assessment by the end of 2015Q3. Most of them (95.0%) were referred to GPC-employed health professionals. Among participants, females (the proportion of females among health check participants was 59.0%) and older (the proportions of 18–44-, 45–64-, and 65+-year-old participants were 39.2%, 37.5%, and 23.3%, respectively) subjects were overrepresented.

The DPR in Hungary was 69.3% in the first quarter of 2012 and decreased to 60.8% by the third quarter of 2015. This reduction was observed in each studied socioeconomic stratum and in each studied ATC group. All of these reductions proved to be statistically significant using a chi-squared test. The DPR difference between males and females was negligible in both study periods. Being over 65 proved to positively influence the DPR, while the DPR was minimal among middle-aged patients and the DPR was higher among patients with exemption certificates than among those without exemption certificates in the studied periods. The highest DPR was observed for anti-infectives for systemic use (DPR_2012Q1_ = 80.1%, DPR_2015Q3_ = 76.1%). The minimum DPR was registered for the group of various drugs in 2012Q1 (DPR = 57.6%) and for cardiovascular system drugs in 2015Q3 (DPR = 55.3%). Differences between DPRs before and after the intervention were statistically significant across sociodemographic and ATC groups (Table 2).

The registered number of prescriptions and dispensed prescriptions in the intervention population was 134,470 and 98,213, respectively, for 2012Q1. The observed DPR before the intervention was 73.0%. The influence of sex and age on the DPR was moderate, while the difference between DPRs for patients eligible and not eligible for an exemption certificate was statistically significant. The ATC-specific DPR variability by ATC group was also statistically significant. The highest and lowest DPRs were observed for systemic hormonal preparations (81.8%) and cardiovascular system drugs (68.8%), respectively (Table 3).

The DPR for after the intervention period was 68.7% (133,689 prescriptions; 91,881 dispensed). The impact of sex and the eligibility for an exemption certificate showed no change. However, the age group effects widened considerably. The ranges of the ATC-specific DPR variability and the extreme groups did not change (from the lowest DPR of 62.0% for cardiovascular system drugs to the highest DPR of 81.2% for systemic hormonal preparations). 

According to the sDPRs, medication adherence was slightly higher in the intervention area than in Hungary in general for each ATC group. The before-intervention sDPR of 1.042 increased to 1.108 by the 2015Q3 period. This change in sDPR proved to be significant (RDR = 1.064; 95% CI: 1.054–1.073; *p* < 0.001) and corresponded to 5033.2 excess dispensed prescriptions. In the ATC groups, the most significant changes were observed for cardiovascular system drugs (RDR = 1.062; 95% CI: 1.048–1.077; *p* < 0.001) and for alimentary tract and metabolism-specific drugs (RDR = 1.072; 95% CI: 1.049–1.097; *p* < 0.001), corresponding to 2143.5 and 1001.2 excess dispensed prescriptions, respectively. Significant improvement was observed for nervous system drugs (RDR = 1.082; 95% CI: 1.047–1.118; *p* < 0.001), blood and blood-forming organ drugs (RDR = 1.077; 95% CI: 1.044–1.111; *p* < 0.001), and musculoskeletal (RDR = 1.041; 95% CI: 1.010–1.074; *p* = 0.010) drugs (Table 4).

## 4. Discussion 

### 4.1. Main Findings

In this study, we estimated the impact of a PHC-level, multifaceted intervention on the enhancement of primary adherence to prescribed medications. The overall increase in the age-, sex-, and exemption certificate eligibility-standardized dispensed percentage of prescribed medications was 6%. Indeed, we observed that the percentage of drugs actually dispensed declined over time in both the intervention area and in Hungary. However, the decline was sharper in Hungary in general compared to that in the intervention area. This improvement was a secondary impact of activities conducted within the project, which aimed primarily to improve health outcomes through the integration of new primary, secondary, and tertiary preventive services into the PHC system. The achieved improvement of 6% was somewhat modest, but it fit within the range of 4% to 11% reported in other studies [10,63]. It seems probable that completing the GPC intervention with the targeted training modalities would further enhance these achievements.

The most important impacts were observed for cardiovascular system drugs (42.6% excess of dispensed prescriptions) and alimentary tract and metabolism-specific drugs (19.9% excess of dispensed prescriptions), which showed that individuals with diseases of higher prevalence and demand on the PHC workforce were the most responsive to the intervention. This finding suggests that more intensive care for patients plays a key role in improving primary adherence. The nonspecific nature of the intervention with respect to medication adherence is supported by the fact that significant improvements with smaller impacts were also observed in the case of nervous system drugs (10.7% of excess dispensed prescriptions), blood and blood-forming organ drugs (9.8% of excess dispensed prescriptions), and musculoskeletal drugs (7.4% of excess dispensed prescriptions). 

These findings are in good concordance with the well-known fact that the patient–provider relationship is an important predictor of adherence [64,65]. Improved collaboration entails mutual understanding and creates trust [66,67,68,69] that affects patients’ beliefs about medications and enhances adherence [70,71,72]. In addition, this improved patient–physician relationship may give patients a more active role in the therapeutic process and in decision-making, taking into account patient perspectives and eventually leading to enhanced adherence [73].

An important condition of the study was that the DPRs from 2012Q1 and those from the last available period of 2015Q3 showed remarkable differences in Hungary. The causes of the national reference data change—controlled for by the study design—were not investigated, but they may be attributable to seasonal effects, taking into consideration that 2012Q1 included January, February, and March while 2015Q3 included July, August, and September. They may also be attributable to a reduced number of patients with an exemption certificate.

### 4.2. Implications

In fact, there was no training in the GPC intervention to improve the healthcare professionals’ ability to educate patients, support patients psychologically, or organize family support in order to improve patients’ medication adherence. Our results confirm that an improvement in client–physician cooperation (e.g., by integrating missing preventive services into a PHC) leads to improved medication adherence without intentionally targeting this change. Therefore (not debating the importance of targeted programs to medication adherence [63]), our observations suggest that medication adherence should be systematically checked by DPR-like indicators in the case of any PHC development project, since a positive impact on medication adherence can be a secondary effect of any intervention that influences cooperation between patients and health professionals [74]. It seems that DPR-like measures can be used as indicators for the effectiveness of cooperation between clients and health professionals.

Exploitation of this opportunity seems to be especially important, since medical doctors have no proper information on medication nonadherence, and they underestimate the frequency of it [75,76]. Additionally, medical doctors express their need for feedback, and they would like to know whether their practice needs to be modified or not [77].

Furthermore, currently available information technology (IT) could produce indicators for routine monitoring without exceptional costs, as demonstrated by our investigation. This could support general benchmarking activities.

### 4.3. Strengths and Limitations

This study was a before–after analysis. Therefore, potential confounding factors (socioeconomic status of participants, symptom severity and disability among participants, available therapeutic tools) that were not taken into account in the statistical analysis could be considered constants. Using national reference data, the design of the study made it possible to control for factors that could potentially influence the DPR and change throughout the country without identifying and quantifying them. Without this design element, the study could not have been informative regarding the effect of GPCs on medication adherence.

Although dispensing medications is an initial step in the therapeutic process, it cannot fully determine whether the patient really uses the medication and follows physician instructions properly.

A person-level linkage between the actually used extra services of a GPC, drug consumption, and data on prescribed and dispensed medications was not feasible in our investigation. Considering that more than half of the target population participated in the health status assessment by the end of the study period and that practically all of these participants were involved in GPC-specific extra care, approximately half of the target population produced improvement in the intervention area-specific DPR. The observed impact on medication adherence was an underestimation of the real influence of a GPC. Consequently, this validity issue did not jeopardize the conclusion on the effectiveness of the intervention.

Because we carried out indirect standardization according to the sociodemographic status of subjects, where the demographic strata-specific redemption could not be quantified, our analysis could not identify the strata, which were responsible for the 6% increase in the redemption intention. Furthermore, mechanisms eliciting the reported enhancement of adherence were not studied in our investigation, since adherence enhancement was not among the program’s objectives and the monitoring did not cover this question. Additional investigations are needed to identify the beneficial changes.

## 5. Conclusions

Altogether, our observations demonstrate that the integration of preventive services into a PHC (by ensuring the necessary capacity and elaborating on new protocols) is beneficial for medication adherence, especially among patients with cardiovascular, alimentary tract, and metabolic disorders. The 6% enhancement in the percentage of drugs actually dispensed that was achieved in our GPC-based intervention without any specific medication adherence-increasing activities corresponded to, but did not exceed, the published range of achievements of targeted and multifaceted adherence-increasing interventions. 

Considering the high impact and the multifactorial (but not properly explored) nature of medication adherence, monitoring the percentage of drugs actually dispensed seems to be a rational element of PHC-oriented intervention evaluation frames. Moreover, this indicator should be used in general monitoring of PHCs to support benchmarking. Our findings emphasize that the WHO recommendation for applying the percentage of drugs actually dispensed as a routine indicator should be considered more seriously: the measure of “percentage of drugs actually dispensed” could be a useful indicator of the effectiveness of client–PHC provider collaboration. 

## Figures and Tables

**Table 1 ijerph-16-03797-t001:** Demographic composition of the population in the intervention area and Hungary.

Patient Characteristics		Intervention Area % (*N*)	Hungary % (*N*)	*p*-Value *
First quarter of 2012				
Age group (years)	18–44	46.1% (15,265)	46.5% (3,667,334)	0.340
	45–64	33.2% (10,973)	33.0% (2,602,749)	
	65 and above	20.7% (6863)	20.5% (1,616,579)	
Sex	Male	47.9% (15,855)	46.7% (3,679,137)	<0.001
	Female	52.1% (17,246)	53.3% (4,207,525)	
Exemption certificate	Yes	5.8% (1933)	3.2% (251,027)	<0.001
	No	94.2% (31,168)	96.8% (7,635,635)	
All together		100% (33,101)	100% (7,886,662)	-
Third quarter of 2015				
Age group (years)	18–44	45.4% (14,690)	44.6% (3,451,254)	<0.001
	45–64	32.5% (10,499)	33.3% (2,578,267)	
	65 and above	22.1% (7133)	22.2% (1,715,591)	
Sex	Male	47.8% (15,449)	46.7% (3,619,811)	<0.001
	Female	52.2% (16,873)	53.3% (4,125,301)	
Exemption certificate	Yes	5.3% (1718)	2.5% (194,678)	<0.001
	No	94.7% (30,604)	97.5% (7,550,434)	
All together		100% (32,322)	100% (7,745,112)	-

* Chi-squared test.

**Table 2 ijerph-16-03797-t002:** The proportion of dispensed medications prescribed by general practitioners (GPs) in Hungary in the first quarter of 2012 (before-intervention) and in the third quarter of 2015 (after the program implementation) by patient sociodemographic characteristic and anatomical–therapeutic chemical classification (ATC) drug group.

Patient Characteristics and ATC Groups		Prescriptions Before Intervention			Prescriptions After Intervention			
		Written	Dispensed	Dispensed Ratio (%)	Written	Dispensed	Dispensed Ratio (%)	*p*-Value *
Age group (years)	18–44	2,879,000	1,952,263	67.8%	2,525,076	1,529,643	60.6%	<0.001
	45–64	11,732,996	7,889,604	67.2%	10,874,787	6,369,987	58.6%	<0.001
	65 and above	15,190,426	10,822,025	71.2%	14,899,896	9,304,872	62.4%	<0.001
Sex	Male	11,689,243	8,051,849	68.9%	11,158,720	6,711,249	60.1%	<0.001
	Female	18,113,179	12,612,043	69.6%	17,141,039	10,493,253	61.2%	<0.001
Exemption certificate	Yes	3,499,275	2,804,373	80.1%	2,709,909	2,054,121	75.8%	<0.001
	No	26,303,147	17,859,519	67.9%	25,589,850	15,150,381	59.2%	<0.001
Alimentary tract and metabolism		4,831,608	3,504,498	72.5%	4,596,768	3,042,485	66.2%	<0.001
Blood and blood-forming organs		2,187,096	1,636,426	74.8%	1,970,831	1,333,257	67.6%	<0.001
Cardiovascular system		15,311,478	10,057,565	65.7%	14,642,073	8,094,617	55.3%	<0.001
Dermatologic		289,326	189,229	65.4%	300,258	176,043	58.6%	<0.001
Genitourinary system and sex hormones		210,643	152,318	72.3%	219,692	143,770	65.4%	<0.001
Systemic hormonal preparations		323,519	244,649	75.6%	354,554	259,049	73.1%	<0.001
Anti-infectives for systemic use		1,208,603	968,386	80.1%	666,892	507,756	76.1%	<0.001
Musculoskeletal system		1,851,092	1,324,670	71.6%	2,093,343	1,389,095	66.4%	<0.001
Nervous system		1,921,312	1,419,382	73.9%	1,845,835	1,255,821	68.0%	<0.001
Respiratory system		1,488,397	1,042,395	70%	1,448,618	901,644	62.2%	<0.001
Sensory organs		128,978	95,371	73.9%	97,004	63,204	65.2%	<0.001
Various		50,370	29,003	57.6%	63,891	37,761	59.1%	<0.001
All together		29,802,422	20,663,892	69.3%	28,299,759	17,204,502	60.8%	<0.001

* Chi-squared test.

**Table 3 ijerph-16-03797-t003:** The proportion of dispensed medications prescribed by GPs in the intervention area before and after the program implementation by patient sociodemographic characteristic and ATC drug group.

Patient Characteristics and ATC Groups		Prescriptions Before Intervention			Prescriptions After Intervention			Change of Dispensed Ratio
		Written	Dispensed	Dispensed Ratio (%)	Written	Dispensed	Dispensed Ratio (%)	*p*-Value *
Age group (years)	18–44	15,808	11,475	72.6%	15,670	10,425	66.5%	<0.001
	45–64	55,673	40,197	72.2%	54,909	36,756	66.9%	<0.001
	65 and above	62,989	46,541	73.9%	63,110	44,700	70.8%	<0.001
Sex	Male	51,818	37,999	73.3%	51,349	35,103	68.4%	<0.001
	Female	82,652	60,214	72.9%	82,340	56,778	69.0%	<0.001
Exemption certificate	Yes	24,372	20,656	84.8%	22,505	18,617	82.7%	0.077
	No	110,098	77,557	70.4%	111,184	73,264	65.9%	<0.001
Alimentary tract and metabolism		20,238	15,326	75.7%	20,536	15,336	74.7%	0.356
Blood and blood-forming organs		10,498	8209	78.2%	9908	7569	76.4%	0.272
Cardiovascular system		64,495	44,383	68.8%	65,030	40,319	62.0%	0.000
Dermatologic		1524	1141	74.9%	1695	1189	70.1%	0.231
Genitourinary system and sex hormones		835	661	79.2%	1098	834	76.0%	0.552
Systemic hormonal preparations		1662	1360	81.8%	1796	1459	81.2%	0.886
Anti-infectives for systemic use		6392	5212	81.5%	3616	2906	80.4%	0.641
Musculoskeletal system		10,083	7627	75.6%	11,773	8679	73.7%	0.215
Nervous system		9192	7133	77.6%	9408	7302	77.6%	0.993
Respiratory system		8667	6530	75.3%	8052	5767	71.6%	0.033
Sensory organs		580	445	76.7%	535	369	69.0%	0.249
Various		304	186	61.2%	242	152	62.8%	0.851
All together		134,470	98,213	73.0%	133,689	91,881	68.7%	<0.001

* Chi-squared test to compare dispensed ratio before and after the intervention.

**Table 4 ijerph-16-03797-t004:** Changes in standardized dispensed-to-prescribed ratio (sDPR) in the intervention area and the number of excess dispensed prescriptions attributable to the program’s implementation (stratified by ATC drug group).

ATC Group	Before Intervention		After Intervention		Change of sDPR		
	sDPR (O/E) *	Excess Dispensing	sDPR (O/E)	Excess Dispensing	RR ** (95% CI)	Excess Dispensing	*p*-Value
Alimentary tract and metabolism	1.035 (15,326/14,809.5)	516.5	1.11 (15,336/13,818.3)	1517.7	1.072 (1.049–1.097)	1001.2	<0.001
Blood and blood-forming organs	1.042 (8209/7878.5)	330.5	1.122 (7569/6745.6)	823.4	1.077 (1.044–1.111)	492.9	<0.001
Cardiovascular system	1.035 (44,383/42,880.5)	1502.5	1.099 (40,319/36,673)	3646	1.062 (1.048–1.077)	2143.5	<0.001
Dermatologic	1.134 (1141/1006.4)	134.6	1.168 (1189/1017.7)	171.3	1.031 (0.950–1.118)	36.7	0.468
Genitourinary system and sex hormones	1.087 (661/608)	53	1.154 (834/722.4)	111.6	1.062 (0.959–1.176)	58.6	0.249
Systemic hormonal preparations *	1.081 (1360/1258.1)	101.9	1.108 (1459/1316.8)	142.2	1.025 (0.952–1.104)	40.3	0.513
Anti-infectives for systemic use	1.011 (5212/5157.1)	54.9	1.052 (2906/2763.2)	142.8	1.041 (0.994–1.089)	87.9	0.086
Musculoskeletal system	1.042 (7627/7320.2)	306.8	1.085 (8679/7999.9)	679.1	1.041 (1.010–1.074)	372.3	0.010
Nervous system	1.045 (7133/6828.7)	304.3	1.13 (7302/6461.4)	840.6	1.082 (1.047–1.118)	536.4	<0.001
Respiratory system	1.147 (6530/5693.2)	836.8	1.128 (5767/5113.5)	653.5	0.983 (0.949–1.019)	-183.4	0.351
Sensory organs	1.031 (445/431.6)	13.4	1.048 (369/352.1)	16.9	1.017 (0.886–1.167)	3.6	0.817
Various	1.057 (186/176.1)	9.9	1.049 (152/144.9)	7.1	0.993 (0.801–1.230)	-2.9	0.950
All together	1.042 (98,213/94,275.9)	3937.1	1.108 (91,881/82,910.7)	8970.3	1.064 (1.054–1.073)	5033.2	<0.001

* (O/E): observed and expected number of dispensed drugs. ** RR = sDPR_after_/sDPR_before._

## Data Availability

The datasets used and/or analyzed during the current study are available from the corresponding author upon reasonable request.

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
