# Peer review of "Enhancing Primary Adherence to Prescribed Medications through an Organized Health Status Assessment-Based Extension of Primary Healthcare Services"

_ijerph, 2019, doi:10.3390/ijerph16203797_

Round 1
Reviewer 1 Report
Thank you for your submission on this topic.
The aim of this study was to demonstrate whether the GPC operation could increase the percentage of drugs actually dispensed. You demonstrated that in your statistical results. I am not sure you can claim that this study enhanced adherence to prescribed drugs as the title suggests. Adherence not only includes the dispensing of drugs but if the patient is taking the prescribed medications as directed. Using a marker such as blood pressure readings would have been a good measurement of adherence but since the data was in aggregate you cannot claim that adherence was improved or enhanced; you can only comment on the number of prescriptions dispensed.
Methods: Statistical analysis- Line 122 and 123-how can you measure adherence to prescribed medications if you only measured if the prescription was dispensed and did not measure the actual taking of the prescribed medication by looking at refill records for on-time refills, counting of pills or monitoring parameters of improved symptoms (i.e BP reductions) or if the prescriptions were appropriate.
Results: Table 1- some figures have a comma and other have a decimal point in the percentages reported- please use a consistent method for reporting percentages.
Line 154 avoid using terms as "highly" significant. Just state results were statistically significant. Line 156 ?should that be DPR was minimal among middle-aged patients"? It does not flow well as currently written. Report p-values in the text as well for results. Line 169 just use statistically significant. Avoid adjectives such as remarkable, considerable- use scientific terms in describing results. Lines 183-190 add p-values to the data reports in the text along with the RDR and 95% CI.
References: many need to be corrected. Some use the the correct abbreviations many others do not- they are mixed formats used. Reference 55 is just a viewpoint, is reference 56 from a web site? please add website address and date accessed. I could not access reference 57
Author Response
Dear Reviewer-1,
Please find enclosed the revised (manuscript ID: ijerph- 600867) manuscript „Enhancing primary adherence to prescribed medications by an organized health status assessment-based extension of primary health care services”, by Nouh Harsha, et al., to be submitted as an article to the International Journal of Environmental Research and Public Health for consideration of publication.
Thank you very much for the careful review of our manuscript. The corresponding changes and refinements made in the revised paper are summarized in our response after considering each of your suggestion. The changes in the manuscript are shown with revision marks.
Answers along with the modifications we made are the following (comments of yours are in capitals):
1.
THE AIM OF THIS STUDY WAS TO DEMONSTRATE WHETHER THE GPC OPERATION COULD INCREASE THE PERCENTAGE OF DRUGS ACTUALLY DISPENSED. YOU DEMONSTRATED THAT IN YOUR STATISTICAL RESULTS. I AM NOT SURE YOU CAN CLAIM THAT THIS STUDY ENHANCED ADHERENCE TO PRESCRIBED DRUGS AS THE TITLE SUGGESTS. ADHERENCE NOT ONLY INCLUDES THE DISPENSING OF DRUGS BUT IF THE PATIENT IS TAKING THE PRESCRIBED MEDICATIONS AS DIRECTED. USING A MARKER SUCH AS BLOOD PRESSURE READINGS WOULD HAVE BEEN A GOOD MEASUREMENT OF ADHERENCE BUT SINCE THE DATA WAS IN AGGREGATE YOU CANNOT CLAIM THAT ADHERENCE WAS IMPROVED OR ENHANCED; YOU CAN ONLY COMMENT ON THE NUMBER OF PRESCRIPTIONS DISPENSED.
In line with the published literature, there are two types of nonadherence to prescribed medications: primary and secondary. The primary nonadherence occurs when patient does not dispense the new prescriptions written by their health care providers from the beginning of the treatment course and this is the one we are interested in this paper. On the other hand, secondary nonadherence occurs in patients who do not use their medication according to the guidelines and recommendations of the health care providers (such as dose frequency and duration) and this can be checked through various methods including therapeutic outcome like BP measurement or taking a blood sample to ensure the actual ingestion of the medicines.
Our data support the investigation of the first type of nonadherence that is primary nonadherence which means in other words dispensing of drugs. To clarify this issue, we have inserted the word “primary” to the title of the study (Line 2-3) and into the section of Objectives (Line 87):
Original title:
Enhancing adherence to prescribed medication use by an organized health status assessment-based extension of primary health care services
Modified title:
Enhancing primary adherence to prescribed medications by an organized health status assessment-based extension of primary health care services
Original sentence in Objectives:
Our study aimed to demonstrate whether the GPC operation, which aimed to improve the general effectiveness of PHC, could increase the percentage of drugs actually dispensed, thereby reflecting improvement in the quality of patient-GP collaboration.
Modified sentence in Objectives:
Our study aimed to demonstrate whether the GPC operation, which aimed to improve the general effectiveness of PHC, could increase the primary adherence, the percentage of drugs actually dispensed, thereby reflecting improvement in the quality of patient-GP collaboration.
2.
METHODS: STATISTICAL ANALYSIS - LINE 122 AND 123 - HOW CAN YOU MEASURE ADHERENCE TO PRESCRIBED MEDICATIONS IF YOU ONLY MEASURED IF THE PRESCRIPTION WAS DISPENSED AND DID NOT MEASURE THE ACTUAL TAKING OF THE PRESCRIBED MEDICATION BY LOOKING AT REFILL RECORDS FOR ON-TIME REFILLS, COUNTING OF PILLS OR MONITORING PARAMETERS OF IMPROVED SYMPTOMS (I.E. BP REDUCTIONS) OR IF THE PRESCRIPTIONS WERE APPROPRIATE.
We tried to clarify this point by the former response. Further, we replaced also the “adherence” with “primary adherence” in line 128:
Original text:
We used DPR as a key indicator of patient care to assess adherence to prescribed medications.
Modified text:
We used DPR as a key indicator of patient care to assess primary adherence to prescribed medications.
3.
RESULTS: TABLE 1 - SOME FIGURES HAVE A COMMA AND OTHER HAVE A DECIMAL POINT IN THE PERCENTAGES REPORTED- PLEASE USE A CONSISTENT METHOD FOR REPORTING PERCENTAGES.
Table 1 has been checked and fixed.
4.
LINE 154 AVOID USING TERMS AS "HIGHLY" SIGNIFICANT. JUST STATE RESULTS WERE STATISTICALLY SIGNIFICANT.
We removed the word “highly”. (Line 162)
Original text
All of these reductions proved to be highly significant by chi-square test.
Modified text
All of these reductions were proved to be statistically significant by chi-square test.
5.
LINE 156: SHOULD THAT BE DPR WAS MINIMAL AMONG MIDDLE-AGED PATIENTS? IT DOES NOT FLOW WELL AS CURRENTLY WRITTEN. REPORT P-VALUES IN THE TEXT AS WELL FOR RESULTS.
We have rephrased the sentence and fixed this point. (Line 163-165)
Original text:
Age above 65 proved to positively influence the DPR, while DPR showed a minimum among middle-aged patients.
Modified text:
Age above 65 proved to positively influence the DPR, while DPR was minimal among middle-aged patients.
Further, we added P values from chi-square test to table 2 with a subscript to declare that chi-square test was applied, and we added a sentence to the text (Line 169-170):
Differences between DPRs before and after intervention were statistically significant across sociodemographic and ATC groups.
6.
LINE 169 JUST USE STATISTICALLY SIGNIFICANT. AVOID ADJECTIVES SUCH AS REMARKABLE, CONSIDERABLE - USE SCIENTIFIC TERMS IN DESCRIBING RESULTS.
Thank you. Noted and corrected (Line 179-181).
Original text
The influence of sex and age on DPR was moderate, while the difference between DPR for patients eligible and not eligible for an exemption certificate was remarkable. The ATC-specific DPR variability by ATC groups was considerable.
Modified text
The influence of sex and age on DPR was moderate, while the difference between DPR for patients eligible and not eligible for an exemption certificate was statistically significant. The ATC-specific DPR variability by ATC groups was also statistically significant.
7.
LINES 183-190 ADD P-VALUES TO THE DATA REPORTS IN THE TEXT ALONG WITH THE RDR AND 95% CI.
We added p values to the table 4 and in the text as well (line 192-201).
Original text
This change in sDPR proved to be significant (RDR=1.064; 95%CI: 1.054-1.073) and corresponded to 5033.2 excess dispensed prescriptions. By ATC group, the most significant changes were observed for cardiovascular system drugs (RDR=1.062; 95%CI: 1.048-1.077) and for alimentary tract and metabolism-specific drugs (RDR=1.072; 95%CI: 1.049-1.097) and corresponded to 2143.5 and 1001.2 excess dispensed prescriptions, respectively. Significant improvement was observed for nervous system drugs (RDR=1.082; 95%CI: 1.047-1.118), blood and blood-forming organ drugs (RDR=1.077; 95%CI: 1.044-1.111), and musculoskeletal (RDR=1.041; 95%CI: 1.010-1.074) drugs.
Modified text
This change in sDPR proved to be significant (RDR=1.064; 95%CI: 1.054-1.073, p<0.001) and corresponded to 5033.2 excess dispensed prescriptions. By ATC group, the most significant changes were observed for cardiovascular system drugs (RDR=1.062; 95%CI: 1.048-1.077, p<0.001) and for alimentary tract and metabolism-specific drugs (RDR=1.072; 95%CI: 1.049-1.097, p<0.001) and corresponded to 2143.5 and 1001.2 excess dispensed prescriptions, respectively. Significant improvement was observed for nervous system drugs (RDR=1.082; 95%CI: 1.047-1.118, p<0.001), blood and blood-forming organ drugs (RDR=1.077; 95%CI: 1.044-1.111, p<0.001), and musculoskeletal (RDR=1.041; 95%CI: 1.010-1.074, p=0.010) drugs.
8.
REFERENCES: MANY NEED TO BE CORRECTED. SOME USE THE CORRECT ABBREVIATIONS MANY OTHERS DO NOT- THEY ARE MIXED FORMATS USED. REFERENCE 55 IS JUST A VIEWPOINT, IS REFERENCE 56 FROM A WEB SITE? PLEASE ADD WEBSITE ADDRESS AND DATE ACCESSED. I COULD NOT ACCESS REFERENCE 57
We have double checked and corrected the reference list to ensure consistency. In addition, we added a new reference (56) that can be accessed instead of the old reference 56.
Sincerely yours,
Janos Sandor on behalf of the authors

Reviewer 2 Report
This manuscript describes secondary outcomes from a primary care preventive healthcare intervention. In general it is clearly presented and its findings are of interest. Some specific issues the authors could consider are outlined below.
It would be helpful to readers who are not familiar with the Hungarian healthcare system briefly to explain the exemption certificate – presumably reflecting a subsidized system of healthcare based on socioeconomic status?
Limited information has been provided on the nature of the preventive healthcare measures comprising the intervention. Was there an emphasis on diet and lifestyle issues that could have involved a focus on their cardiovascular implications, for example, and hence increased the adherence to medication in this class?
L149: It is stated that more than half of the target community participated in an organized healthcare assessment during the study period, most of whom were referred to GPC-employed health professionals. Are there any data on the demographic profile of the participants (as opposed to the overall target group), indicating how representative they may have been?
It would be helpful for Table 2 to include p-values (similar to Table 3).
Table 4 refers to “SDR” in the heading for the last column, and in the footnote to the table. Should this be sDPR in each case?
L214-217: The discussion refers to the potential that improved prescriber-patient relationships may have enhanced adherence. Rather than this simply representing adherence to existing prescriptions, the authors may also consider the possibility that greater prescriber-patient engagement may have led to a more collaborative decision-making process yielding amended prescriptions more in keeping with patients’ needs/perspectives.
L219-221: It is stated that the causes of the national reference data change were not investigated but “they can be attributed to seasonal effects”. This statement seems stronger than is justified by the data. In the absence of further investigation, a more cautious statement such as “they may be attributable to seasonal effects” would be preferable.
In the limitations section, it would be appropriate to acknowledge that dispensing records are an imperfect measure of adherence; medication possession commonly does not translate into medication administration, as reflected in the references cited by the authors at L37-38.
L267: The authors refer to a “6% enhancement for the percentage of drugs actually dispensed”. This statement should be clarified since the simple “percentage of drugs actually dispensed” declined over the study period; however since the decline was not as steep in the intervention group as in the general population, the relative ratio improved.
Overall, the manuscript is well written, although a small number of sections could be more clearly expressed – notably Section 2.4 (L135-138) and L238-239 – to communicate their meaning more effectively. The column headings in Table 3 could also be formatted (capitalization) in a manner that is consistent with Table 2. The formatting of references also requires attention, especially as regards capitalization of journal titles.
Author Response
Dear Reviewer-2,
Please find enclosed the revised (manuscript ID: ijerph- 600867) manuscript „Enhancing primary adherence to prescribed medications by an organized health status assessment-based extension of primary health care services”, by Nouh Harsha, et al., to be submitted as an article to the International Journal of Environmental Research and Public Health for consideration of publication.
Thank you very much for the careful review of our manuscript. The corresponding changes and refinements made in the revised paper are summarized in our response after considering each of your suggestion. The changes in the manuscript are shown with revision marks.
Answers along with the modifications we made are the following (comments of yours are in capitals):
1.
THIS MANUSCRIPT DESCRIBES SECONDARY OUTCOMES FROM A PRIMARY CARE PREVENTIVE HEALTHCARE INTERVENTION. IN GENERAL IT IS CLEARLY PRESENTED AND ITS FINDINGS ARE OF INTEREST. SOME SPECIFIC ISSUES THE AUTHORS COULD CONSIDER ARE OUTLINED BELOW.
IT WOULD BE HELPFUL TO READERS WHO ARE NOT FAMILIAR WITH THE HUNGARIAN HEALTHCARE SYSTEM BRIEFLY TO EXPLAIN THE EXEMPTION CERTIFICATE – PRESUMABLY REFLECTING A SUBSIDIZED SYSTEM OF HEALTHCARE BASED ON SOCIOECONOMIC STATUS?
Exemption certificate is a certificate issued by the National Health Insurance Fund to the socioeconomically disadvantaged patients who require long-term care after being diagnosed by the relevant GP and approved to deserve exemption by the relevant municipality. An explanation has been added to sentence (Line 117-119).
Original text:
… eligibility for an exemption certificate.
Completed text:
… eligibility for an exemption certificate (by which socioeconomically disadvantaged patients who require long-term care can get drugs and medical devices free of charges).
2.
LIMITED INFORMATION HAS BEEN PROVIDED ON THE NATURE OF THE PREVENTIVE HEALTHCARE MEASURES COMPRISING THE INTERVENTION. WAS THERE AN EMPHASIS ON DIET AND LIFESTYLE ISSUES THAT COULD HAVE INVOLVED A FOCUS ON THEIR CARDIOVASCULAR IMPLICATIONS, FOR EXAMPLE, AND HENCE INCREASED THE ADHERENCE TO MEDICATION IN THIS CLASS?
We extended the explanation by completing the original sentences (Line 105-106).
Inserted text:
Further, new services of the programme which were not available formerly at the level of PHC included the following:
3.
L149: IT IS STATED THAT MORE THAN HALF OF THE TARGET COMMUNITY PARTICIPATED IN AN ORGANIZED HEALTHCARE ASSESSMENT DURING THE STUDY PERIOD, MOST OF WHOM WERE REFERRED TO GPC-EMPLOYED HEALTH PROFESSIONALS. ARE THERE ANY DATA ON THE DEMOGRAPHIC PROFILE OF THE PARTICIPANTS (AS OPPOSED TO THE OVERALL TARGET GROUP), INDICATING HOW REPRESENTATIVE THEY MAY HAVE BEEN?
We have added some points on the demographic structure of the participants’ group (Line 157-159)
New text:
Among participants, females (proportion of female among health check participants was 59.0%) and older (proportion of 18-44, 45-64, and 65+ years old participants were 39.2%, 37.5%, and 23.3%, respectively) subjects were overrepresented.
4.
IT WOULD BE HELPFUL FOR TABLE 2 TO INCLUDE P-VALUES (SIMILAR TO TABLE 3).
We added P values from chi-square test to table 2 with a subscript to declare that chi-square test was applied, and we added a sentence to the text (Line 169-170):
Differences between DPRs before and after intervention were statistically significant across sociodemographic and ATC groups.
5.
TABLE 4 REFERS TO “SDR” IN THE HEADING FOR THE LAST COLUMN, AND IN THE FOOTNOTE TO THE TABLE. SHOULD THIS BE SDPR IN EACH CASE?
Yes, of course. We have fixed. Many thanks.
6.
L214-217: THE DISCUSSION REFERS TO THE POTENTIAL THAT IMPROVED PRESCRIBER-PATIENT RELATIONSHIPS MAY HAVE ENHANCED ADHERENCE. RATHER THAN THIS SIMPLY REPRESENTING ADHERENCE TO EXISTING PRESCRIPTIONS, THE AUTHORS MAY ALSO CONSIDER THE POSSIBILITY THAT GREATER PRESCRIBER-PATIENT ENGAGEMENT MAY HAVE LED TO A MORE COLLABORATIVE DECISION-MAKING PROCESS YIELDING AMENDED PRESCRIPTIONS MORE IN KEEPING WITH PATIENTS’ NEEDS/PERSPECTIVES.
We have included this possibility in the text along with a new reference (73) to support our point of view (Line 231-233).
New text:
In addition, this improved patient-physician relationship may have given patients more active role in the therapeutic process and decision making that takes into account patient perspective, leading eventually to enhance adherence (73).
7.
L219-221: IT IS STATED THAT THE CAUSES OF THE NATIONAL REFERENCE DATA CHANGE WERE NOT INVESTIGATED BUT “THEY CAN BE ATTRIBUTED TO SEASONAL EFFECTS”. THIS STATEMENT SEEMS STRONGER THAN IS JUSTIFIED BY THE DATA. IN THE ABSENCE OF FURTHER INVESTIGATION, A MORE CAUTIOUS STATEMENT SUCH AS “THEY MAY BE ATTRIBUTABLE TO SEASONAL EFFECTS” WOULD BE PREFERABLE.
Thank you. We modified the statement (Line 236-237).
Original text:
The causes of the national reference data change - controlled for by the study design - were not investigated, but they can be attributed to seasonal effects.
Modified text:
The causes of the national reference data change - controlled for by the study design - were not investigated, but they may be attributable to seasonal effects.
8.
IN THE LIMITATIONS SECTION, IT WOULD BE APPROPRIATE TO ACKNOWLEDGE THAT DISPENSING RECORDS ARE AN IMPERFECT MEASURE OF ADHERENCE; MEDICATION POSSESSION COMMONLY DOES NOT TRANSLATE INTO MEDICATION ADMINISTRATION, AS REFLECTED IN THE REFERENCES CITED BY THE AUTHORS AT L37-38.
Yes, dispensing the medication does not guarantee its ingestion by the patient. However, it represents the initial decision of the patient to follow physician instructions and guidelines. We have added a sentence in the limitations section to explain this (Line 269-271).
New added sentence:
Although dispensing medications is the initial step in the therapeutic process, it cannot fully determine whether the patient has really consumed the medication and followed physicians’ instructions properly.
See also line 280-283 please, new text:
Because we carried out indirect standardization according to the socio-demographic status of subjects, where the demographic strata specific redemption could not be quantified, our analysis could not identify the strata which were responsible for the 6% increase of the redemption intention.
9.
L267: THE AUTHORS REFER TO A “6% ENHANCEMENT FOR THE PERCENTAGE OF DRUGS ACTUALLY DISPENSED”. THIS STATEMENT SHOULD BE CLARIFIED SINCE THE SIMPLE “PERCENTAGE OF DRUGS ACTUALLY DISPENSED” DECLINED OVER THE STUDY PERIOD; HOWEVER SINCE THE DECLINE WAS NOT AS STEEP IN THE INTERVENTION GROUP AS IN THE GENERAL POPULATION, THE RELATIVE RATIO IMPROVED.
We tried to clarify this point in the discussion under main findings (Line 210-212).
New text
Indeed, we have observed that the percentage of drugs actually dispensed declined overtime in both the intervention area and in Hungary. However, the decline was sharper in Hungary in general compared to that in the intervention area.
10.
OVERALL, THE MANUSCRIPT IS WELL WRITTEN, ALTHOUGH A SMALL NUMBER OF SECTIONS COULD BE MORE CLEARLY EXPRESSED – NOTABLY SECTION 2.4 (L135-138) AND L238-239 – TO COMMUNICATE THEIR MEANING MORE EFFECTIVELY. THE COLUMN HEADINGS IN TABLE 3 COULD ALSO BE FORMATTED (CAPITALIZATION) IN A MANNER THAT IS CONSISTENT WITH TABLE 2. THE FORMATTING OF REFERENCES ALSO REQUIRES ATTENTION, ESPECIALLY AS REGARDS CAPITALIZATION OF JOURNAL TITLES.
We have modified the above mentioned sections to ensure clarity.
a) to L140-141: The sentence has been improved and paraphrased (Line 140-143):Original text:
While patients’ needs and public health importance were focused by this investigation, …
Modified text:
Although patients’ needs and the public health importance were investigated in this study, …
b) to L 238-239: The sentence has been improved and paraphrased (Line 254-256):Original text:
Additionally, it seems to be feasible that, since medical doctors express their need for feedback in this area, they would like to know whether their practice will be modified.
Modified text:
Additionally, medical doctors express their need for feedback, they would like to know whether their practice needs to be modified or not.
c) to Table 3:Table 3 capitalization is done.
d) to references:References have been checked and fixed.
Sincerely yours,
Janos Sandor on behalf of the authors

Round 2
Reviewer 1 Report
Thank you for your revised manuscript.
I have just a couple of suggestions:
Page 7 of 14 line 207-208 insert the word "primary" to the sentence"...on enhancing adherence to prescribed medications." so it reads "primary adherence" it will match the title and the aim of your study.
Page 8 of 14 line 222-consider adding the word "primary" in the sentence "..key role in improving adherence." so it reads primary adherence.
Reference 16. Add the website address to the reference and the date accessed.
https://www.who.int/chp/knowledge/publications/adherence_report/en/
Reference 23. Add the website address to the reference and the date accessed.
http://apps.who.int/medicinedocs/en/d/Js2289e/
Reference 68. The correct abbreviation for this journal is: Ann Behave Med
You did a lot of work to revise the manuscript-nicely done!